# Effectiveness of Colorectal Cancer (CRC) Screening on All-Cause and CRC-Specific Mortality Reduction: A Systematic Review and Meta-Analysis

**DOI:** 10.3390/cancers15071948

**Published:** 2023-03-24

**Authors:** Senshuang Zheng, Jelle J. A. Schrijvers, Marcel J. W. Greuter, Gürsah Kats-Ugurlu, Wenli Lu, Geertruida H. de Bock

**Affiliations:** 1Medical Center Groningen, Department of Epidemiology, University of Groningen, 9700 RB Groningen, The Netherlands; s.zheng02@umcg.nl (S.Z.);; 2Medical Center Groningen, Department of Radiology, University of Groningen, 9700 RB Groningen, The Netherlands; 3Robotics and Mechatronics (RaM) Group, Technical Medical Centre, Faculty of Electrical Engineering Mathematics and Computer Science, University of Twente, 7522 NH Enschede, The Netherlands; 4Medical Center Groningen, Department of Pathology, University of Groningen, 9700 RB Groningen, The Netherlands; 5Department of Epidemiology and Health Statistics, Tianjin Medical University, Tianjin 300070, China

**Keywords:** colorectal cancer, screening, mortality reduction, simulation models, randomized control trials, meta-analysis

## Abstract

**Simple Summary:**

Colorectal cancer (CRC) screening is one of the most effective measures to prevent CRC resulting in a decrease in CRC mortality. Mortality reduction (MR) from CRC screening was estimated based on large-scale randomized control trials (RCTs) as well as in model studies, as there is a wide range on CRC-specific MR and a lack of estimates of all-cause MR. We found that biennial FIT, gFOBT, single/5-yearly FS, and 10-yearly colonoscopy screenings reduced CRC-specific mortality significantly, and 10-yearly colonoscopy is the most effective with a mortality reduction of 73%. The effectiveness of screening increases at younger screening initiation ages and higher adherences. Our findings also suggest that adherence is an important factor in CRC-specific mortality and is an explanation for discrepancy in thus far published pooled estimates.

**Abstract:**

(1) Background: The aim of this study was to pool and compare all-cause and colorectal cancer (CRC) specific mortality reduction of CRC screening in randomized control trials (RCTs) and simulation models, and to determine factors that influence screening effectiveness. (2) Methods: PubMed, Embase, Web of Science and Cochrane library were searched for eligible studies. Multi-use simulation models or RCTs that compared the mortality of CRC screening with no screening in general population were included. CRC-specific and all-cause mortality rate ratios and 95% confidence intervals were calculated by a bivariate random model. (3) Results: 10 RCTs and 47 model studies were retrieved. The pooled CRC-specific mortality rate ratios in RCTs were 0.88 (0.80, 0.96) and 0.76 (0.68, 0.84) for guaiac-based fecal occult blood tests (gFOBT) and single flexible sigmoidoscopy (FS) screening, respectively. For the model studies, the rate ratios were 0.45 (0.39, 0.51) for biennial fecal immunochemical tests (FIT), 0.31 (0.28, 0.34) for biennial gFOBT, 0.61 (0.53, 0.72) for single FS, 0.27 (0.21, 0.35) for 10-yearly colonoscopy, and 0.35 (0.29, 0.42) for 5-yearly FS. The CRC-specific mortality reduction of gFOBT increased with higher adherence in both studies (RCT: 0.78 (0.68, 0.89) vs. 0.92 (0.87, 0.98), model: 0.30 (0.28, 0.33) vs. 0.92 (0.51, 1.63)). Model studies showed a 0.62–1.1% all-cause mortality reduction with single FS screening. (4) Conclusions: Based on RCTs and model studies, biennial FIT/gFOBT, single and 5-yearly FS, and 10-yearly colonoscopy screening significantly reduces CRC-specific mortality. The model estimates are much higher than in RCTs, because the simulated biennial gFOBT assumes higher adherence. The effectiveness of screening increases at younger screening initiation ages and higher adherences.

## 1. Introduction

The incidence and mortality of colorectal cancer (CRC) accounts for approximately 10% of all cancers worldwide, with an estimated 1.93 million new cases diagnosed and 0.94 million deaths in 2020 [1,2]. The 5-year CRC survival in 2014 was over 60% in high-income countries, and less than 50% in South American and Asian countries [3,4]. The majority of CRC arises from precursor lesions in the classic pathway with the most common lesions being adenomas and serrated pathways with polypus serrated lesions [4,5]. Usually, it takes 10–15 years for these precursor lesions to progress to CRC [6,7]. CRC screening is one of the most effective measures to prevent CRC resulting in decrease in CRC mortality [4,8]. As such, CRC screening is recommended by the World Health Organization (WHO) and has been implemented in several countries [4,9]. Biannual FIT for people under 75 years of age is the most common screening scenario in countries where population-based CRC screening have been implemented [4].

The mortality reduction (MR), life years gained (LYG) and quality-adjusted life year (QALY) gained from CRC screening were evaluated in large-scaled randomized control trials (RCTs) and in model studies. Five RCTs with guaiac-based fecal occult blood tests (gFOBT) in 765,685 participants and four RCTs with flexible sigmoidoscopy (FS) in 458,022 participants on CRC screening have been reported in the CRC handbook of the International Agency for Research on Cancer [4]. In addition, several models have been widely used to evaluate CRC screening scenarios efficiently and economically [10,11,12,13,14].

Published RCTs and simulation models present a varying range on CRC-specific MR. Four systematic reviews and meta-analyses combined the results of RCTs with gFOBT and FS in intention-to-screen analysis [15,16,17,18]. These reviews included studies that reported relative risks for CRC-specific mortality, ranging from 0.78 to 0.91 using gFOBT and from 0.33 to 0.78 using FS [15,16,17,18]. In the pooled analyses, gFOBT screening leads to a MR of 17% with 95% confidence intervals (95% CI) of 8–25% [17], and that FS results in a MR of 28% (95% CI: 20–35%) [15,16]. The model studies showed 24–79%, 8–84%, 25–56%, 16–94%, and 55–81% CRC-specific MRs on biennial fecal immunochemical tests (FIT), biennial gFOBT, single FS, 10-yearly colonoscopy, and 5-yearly FS screening, respectively [12,13,19,20,21,22,23,24,25,26,27,28,29,30,31,32,33,34,35,36,37,38,39,40,41,42,43]. Adherence rates, adenoma detection rates and dwelling time were used in several model studies in the sensitivity analyses [10,13,19,23,40,41,44].

In conclusion, there is uncertainty over CRC-specific MRs and lack of estimates of all-cause MR in the general population due to CRC screening, and the model studies tend to give higher estimates than RCTs for disease-specific mortalities. Therefore, in this systematic review and meta-analysis, we aim to synthesize and compare the effectiveness of different CRC screening interventions in the general population on all-cause and CRC-specific MR compared with no screening in RCTs and simulation models. In addition, we aim to evaluate the factors that influence screening effectiveness to determine how CRC screening could be improved.

## 2. Materials and Methods

We registered a predefined protocol of this study in the International Prospective Registry of Systematic Reviews (PROSPERO registration number: CRD42021270887). This systematic review and meta-analysis followed the Preferred Reporting Items for Systematic Review and Meta-Analysis Protocols (PRISMA), 2020 statement [45].

### 2.1. Data Sources and Search Strategies

We conducted a systematic literature search in PubMed, Embase, Web of Science and Cochrane library for published RCT studies (1 January 2006 to 31 July 2022) and model studies (1 January 2016 to 31 July 2022) with the following keywords: (“randomized controlled trials”) for RCTs and (“computer simulation” or “models” or “modelling” or “Markov chain”) for simulation models, (“mortality” or “cost-benefit analysis” or “effectiveness” or “life-year”), (“early detection of cancer” or “mass screening” or “fecal immunochemical test” or “fecal occult blood test“ or “colonoscopy” or “sigmoidoscopy”), and (“colorectal cancer” or “bowel cancer” or “colon cancer” or “rectum cancer”). The keywords retrieving RCTs were used for the Cochrane library since this database only includes trial studies. Detailed search strategies for all databases are shown in Appendix A.

### 2.2. Study Selection and Data Extraction

Two reviewers (SZ, JS) independently screened the potentially relevant studies based on eligibility criteria and extracted data from included studies. A study was eligible for inclusion if the following criteria were met: (1) multi-use simulation model or the latest publication of RCT compared commonly used CRC screening scenarios with no screening in general population; (2) original study published in English with outcomes on survival, death number, and CRC-specific or all-cause MR by CRC screening. Studies published as conference abstract, editorial, review, research protocol, study design, or implementation report of screening program without original data were excluded. For a detailed overview of the inclusion and exclusion criteria see Appendix A.

Study information, country/population, screening scenario, adherence to screening, screening time, and follow-up time were extracted for both study types. Screening program, number of participants and person-years of observation in screening and control group, all-cause/CRC death number, and compliance-adjusted outcomes were obtained for RCTs; model used, and all-cause/CRC mortality in screening and no screening scenario for model studies.

### 2.3. Quality Assessment for RCTs and Simulation Models

For the RCT studies, we applied the revised Cochrane risk-of-bias tool for randomized trials (RoB2) [46]. There are five domains in this tool, including randomization process, deviations from intended interventions, missing outcome data, measurement of outcome, and selection of reported results. We classified the risk of bias on each domain and study as “low”, “high”, or “some concerns”.

For the simulation models, we selected the study with the most complete description to evaluate model quality. A qualitative assessment framework included modelling approach, model parameters, transparency of data sources/assumptions, and external validation to assess the overall risk of bias (Appendix A) [47]. When there were two or more items that did not provide a clear description, high risk of bias was assessed; otherwise, low risk. The two reviewers resolved disagreements on study selection, data extraction, and assessments by discussion and further review, or arbitrage by a third author (GdB).

### 2.4. Data Synthesis and Analysis

Data were synthesized and analyzed separately according to screening interventions (gFOBT or FS) in RCT studies. The primary outcomes were all-cause and CRC-specific MR, as measured by calculating the mortality rate ratio between screening and no screening scenarios. We used the original data in a pooled analysis of an intention to screen analysis, which included all individuals as randomized. Because of various follow-up times, observed person-years were extracted to calculate mortality density. RCT studies providing compliance-adjusted results were included in the compliance-adjusted analysis. Rate ratio, hazard ratio, and relative risk were assumed to be similar in the case of large sample sizes and were pooled directly. Regarding the model studies, we summarized data by screening scenarios and synthesized the commonly used global and Dutch scenarios.

Due to differences in the population, trial procedures, model assumptions, and interventions of the included RCTs and simulation models, the expected effect value of these studies were not identical. To ensure that all effects were represented in the pooled effects, and not overly influenced by one study, a bivariate random model was applied to estimate pooled all-cause or CRC-specific mortality rate ratio with 95% CI. The no screening group/scenario was the reference. We then qualitatively synthesized all-cause MRs in model studies.

Meta regression was applied before subgroup analyses to explore possible sources of heterogeneity (I2 ≥ 50%) when there were more than 10 studies/scenarios. Our subgroup analyses followed if heterogeneity existed or when we observed differences in factors affecting screening effectiveness among the pooled studies. We classified subgroups by risk of bias, adherence, screening initiation age, population, model, or screening scenario and assessed publication bias by funnel plots or Egger’s test. A two-sided *p* < 0.05 was considered statistically significant. Statistical analysis was performed using meta package in R 3.6.1 and Review Manager (Version 5.4.1, The Cochrane Collaboration, 2020, London, UK).

## 3. Results

### 3.1. Literature Search

The literature search identified 8051 studies for RCTs and 4695 studies for model studies. A total of 103 RCTs and 162 model studies remained after removing duplicates and reviewing titles and abstracts. We excluded 93 RCTs and 115 model studies following the exclusion criteria. Thus, 10 RCTs and 47 model studies were eligible for inclusion (Figure 1).

### 3.2. RCTs

#### 3.2.1. Quality Assessment

Concerning deviations from intended interventions, five RCTs [48,49,50,51,52] showed some concerns on risk of bias because blind methods were not used, and the other five studies [53,54,55,56,57] were evaluated as high risk because of the lack of blind methods and an appropriate analysis to estimate effect of adhering to intervention. The study of Thiis-Evensen et al. was deemed to have some concerns on randomization process due to small sample size [57]. Overall, five studies were considered as low risk, four studies as some concerns, and one study as high risk of bias (Appendix A).

#### 3.2.2. Study Characteristics

Five RCTs used gFOBT and the other five RCTs used FS as CRC screening interventions (Table 1). There were four European trials and one US trial with gFOBT screening [49,51,52,54,56]. The total number of participants ranged from 46,551 in the US trial to 360,492 in the Finland study. Adherence rates varied from 57.0% to 90.0%. The RCTs with FS screening consisted of four European trials and one US trial [48,50,53,55,57]. Among these trials, the total number of participants ranged from 799 to 170,034. The adherence rates ranged from 57.8% to 86.6%.

#### 3.2.3. Synthesis Results of CRC-Specific and All-Cause MR

In the intention-to-screen analyses, the pooled estimates of all-cause and CRC-specific mortality rate ratios of gFOBT trials were 1.00 (95% CI: 0.99–1.01, *p* = 0.65) and 0.88 (95% CI: 0.80–0.96, *p* = 0.005), and of FS trials 1.02 (95% CI: 0.97–1.06, *p* = 0.46) and 0.76 (95% CI: 0.68–0.84, *p* < 0.001), respectively. There were heterogeneities among the gFOBT studies on the CRC-specific mortality rate ratio (I2 = 59%, *p* = 0.04) and among the FS studies on the all-cause mortality rate ratio (I2 = 88%, *p* < 0.001) (Table 2). The funnel plots do not suggest relevant publication bias (Appendix A).

The pooled estimates of rate ratios in the compliance-adjusted analysis showed that gFOBT screening reduced all-cause mortality by 1% (0.99, 95% CI: 0.97–1.00, *p* = 0.01), CRC–specific mortality by 21% (0.79, 95% CI: 0.68–0.91, *p* = 0.001), and FS screening reduced CRC–specific mortality by 41% (0.59, 95% CI: 0.51–0.70, *p* < 0.001). However, there was no significant reduction in all-cause mortality of FS screening compared with no screening (0.97, 95% CI: 0.91–1.03, *p* = 0.26) (Appendix A).

#### 3.2.4. Subgroup Analysis

Subgroup analysis by adherence showed that there was a significant difference between the ≥70% and <70% adherence groups, and CRC-specific MR by FOBT increased with higher adherence (0.78 (0.68, 0.89) vs. 0.92 (0.87, 0.98), *Psub* = 0.03). For all-cause MR by FS, subgroups of initiation age and risk of bias did not eliminate heterogeneity among studies in intention-to-screen analyses (initiation age: *Psub* = 0.08; risk of bias: *Psub* = 0.19) (Appendix A). In compliance-adjusted analyses, annual gFOBT had a larger significant reduction in CRC-specific mortality than biennial gFOBT (0.65 (0.52, 0.81) vs. 0.84 (0.74, 0.94), *Psub* = 0.045) (Appendix A).

### 3.3. Simulation Models

#### 3.3.1. Study Characteristics and Quality Assessment

Nine simulation models were identified, including three Cancer Intervention and Surveillance Modeling Network (CISNET) CRC models (CRC Simulated Population Model for Incidence and Natural History (CRC-SPIN) [14,24,30,31,42], Simulation Model of Colorectal Cancer (SimCRC) [14,24,30,31,36,42] and MIcrosimulation SCreening ANalysis-Colon (MISCAN-Colon) [14,20,21,22,24,25,29,30,31,35,36,37,38,39,42,58,59,60,61,62,63,64,65,66,67]), adenoma and serrated pathway to colorectal cancer (ASCCA) [12,19,26,27,28,43], Microsimulation-based colon modelling open-source tool (CMOST) [23,41], Colorectal Cancer and Adenoma Incidence and Mortality Microsimulation Model (CRC-AIM) [40,44,68,69], Decision analytic Markov cohort model [32,70,71,72], Multistate Markov model [73,74], and Policy1-Bowel [13,33,34,75]. Six models were assessed as low risk of bias, and three models as high risk (Appendix A).

Five main scenarios were extracted from included studies. In total, 15 studies assessed the effectiveness of biennial FIT screening from 55–75 years [12,19,22,26,27,28,30,31,36,40,43,59,60,62,63], 4 studies biennial gFOBT screening from 45–80 years [25,31,36,40], 6 studies single FS screening from 50–75 years [20,21,24,25,29,42], 7 studies 10-yearly colonoscopy screening from 55–75 years [22,30,31,36,40,43,62], and 4 studies 5-yearly FS screening from 55–75 years [30,31,36,40] (Table 3 and Appendix A). These studies used ASCCA, CRC-AIM, SimCRC, CRC-SPIN, or MISCAN-Colon models, and applied the US, Australian, and European population.

#### 3.3.2. Synthesis Results of CRC-Specific and All-Cause MR

Pooled estimates of CRC-specific mortality rate ratios of scenarios with 55–75 years biennial FIT, 45–80 years biennial gFOBT, 50–75 years single FS, 55–75 years 10-yearly colonoscopy, and 55–75 years 5-yearly FS were 0.45 (95% CI: 0.39–0.51, *p* < 0.001), 0.31 (95% CI: 0.28–0.34, *p* < 0.001), 0.61 (95% CI: 0.53–0.72, *p* < 0.001), 0.27 (95% CI: 0.21–0.35, *p* < 0.001), and 0.35 (95% CI: 0.29–0.42, *p* < 0.001), respectively (Figure 2, Figure 3, Figure 4 and Appendix A). The Egger’s tests indicated that there was a publication bias in model studies on CRC-specific MR except for single FS scenario (Appendix A). The results of screenings from 50 to 75 and 45 to 75 years are shown in Appendix A. The pooled rate ratios of 45–75 years biennial FIT, 10-yearly colonoscopy, and 5-yearly FS were 0.38 (95% CI: 0.32–0.46, *p* < 0.001), 0.17 (95% CI: 0.13–0.24, *p* < 0.001), and 0.27 (95% CI: 0.22–0.33, *p* < 0.001), respectively.

The all-cause MR was presented in two studies. In the one from Norway, the results of the 3% CRC risk population were selected [20,76]. A total of 1.1–1.4% all-cause MR was shown with perfect adherence annual/biennial FIT and single FS/colonoscopy from 50 to 79 years [20]. In the Dutch study, single FS reduced 0.62% of all-cause mortality with 73% adherence [29] (Appendix A).

#### 3.3.3. Subgroup Analysis

There were heterogeneities among studies with 55–75 years 10-yearly colonoscopy on CRC-specific MR (I2 = 55%, *p* < 0.001), and meta regression showed that adherence was the main source for heterogeneity (Appendix A and Appendix A). The pooled estimate of 100% adherence subgroup was 0.21 (95% CI: 0.17–0.27), and of realistic adherence subgroup was 0.70 (95% CI: 0.50–0.97) (*Psub* < 0.001). The results of biennial gFOBT by adherence also indicated a significant difference between 100% and 69% adherence groups (0.30 (0.28, 0.33) vs. 0.92 (0.51, 1.63), *Psub* < 0.001). For single FS, the rate ratio of UK population was lower than that of non-UK population (0.56 (0.48,0.66) vs. 0.70 (0.61,0.80), *Psub* = 0.004) (Appendix A).

#### 3.3.4. The Comparison of Model Studies and RCTs

For CRC-specific MR, pooled estimates of RCTs showed a 12% (4–20%) and 24% (16–32%) reduction in the intention-to-screen analyses, and 21% (9–31%) and 41% (30–49%) reduction in the compliance–adjusted analysis in gFOBT and single FS scenarios, respectively. In model studies, biennial gFOBT presented 69% (66–72%) and single FS 39% (28–47%) reduction. In total, 1% of all–cause MR was found in RCTs with gFOBT when adjusted compliance, and 0.62–1.1% reductions were shown in model studies with single FS.

## 4. Discussion

By systematic selection of RCTs and multiple-use simulation models on all-cause and CRC-specific MR of CRC screening, 10 RCTs and 47 model studies, including 9 simulation models, were retrieved. Our pooled results show that biennial FIT, gFOBT, single/5-yearly FS, and 10-yearly colonoscopy screenings reduced CRC-specific mortality significantly, and 10-yearly colonoscopy is the most effective with a mortality reduction of 73%. Approximately 1% all-cause MR was presented in FIT, gFOBT, and FS scenarios with high adherence or adjusted compliance. Adherence is a crucial factor on the effectiveness of CRC screening, and higher adherence leads to more significant MR. In model studies, younger screening initiation ages were associated with higher MR than older ages.

Although RCTs should be considered as golden standard for evaluation of benefits and cost-effectiveness of screening strategies, trials request large amounts of time and medical resources, and not all potential scenarios can be evaluated. Simulation models eliminate these disadvantages and can assess multiple scenarios [47]. However, our main finding was that the pooled MRs in model studies tended to be higher than in RCTs. An explanation for this overestimation might be uncertainties in model parameters and in assumptions on CRC progression, and most model studies assume ideal parameters and lack external validation [47]. Another possible explanation might be that model studies applied a perfect adherence with lifetime follow-up, which resulted in more than realistic MR [13,77]. Many individuals in screening ages are not screened properly in the real world [77,78].

Our pooled estimates of RCTs showed CRC-specific MR of 12% in gFOBT and 24% in single FS screenings compared to control group. Previous systematic review and meta-analyses reported 12% and 18% MR with gFOBT screening, respectively [79,80]. Others reported 26–28% CRC-specific MR with single FS screening [15,16,80]. The compliance-adjusted analysis showed that FS screening decreased CRC-specific mortality by 41%, which is in accordance with Brenner et al. [15]. Regarding all-cause MR, prior study indicated that single FS and gFOBT screening had little or no reduction in all-cause mortality compared [80]. However, gFOBT screening slightly reduced all-cause mortality by 1% in our results, possibly because only screening participants in intervention group were included, which amplified the effect of screening.

All included scenarios decreased CRC-specific mortality significantly in model studies. Several reviews of model studies concluded that all CRC screening strategies were more effective than no screening [81,82]. In addition, in our pooled estimates, CRC-specific MR of 10-yearly colonoscopy was the highest among all scenarios, while MR of 10-yearly colonoscopy with realistic adherence was not dominant. In general, the adherence of FIT or gFOBT is higher than that of colonoscopy [83]. Thus, although colonoscopy is convinced to have a strong capability in CRC and adenoma screening, the conclusion that the dominance of colonoscopy scenario is not absolute in reality considering the adherence [83,84]. This is consistent with the result of Zhong et al. [83]. Another interesting finding was that biennial gFOBT showed higher MR than biennial FIT. However, gFOBT has lower sensitivity and specificity for CRC than FIT, which results in lower MR than other fecal-based scenarios [31,85,86]. This may be explained by the wider screening age range used with gFOBT, and that all except one study used perfect adherence and lifetime follow-up.

Another finding was that adherence and screening initiation age are crucial factors on effectiveness of CRC screening. Most models included aimed to compare the cost-effectiveness of screening scenarios under optimal conditions. For that, a 100% compliances was assumed. However, in daily screening practice, only part of the invited population will attend CRC screening, which will reduce the screening efficiency [87]. Reported estimates for CRC screening adherence are over 60% in high-income countries in Europe, and generally less than 40% in Eastern European countries [88]. Therefore, the use of the real adherence estimates in simulation models will show more realistic values for the evaluation of screening scenarios. Additionally, the WHO stated that a high adherence is the critical factor for a successful screening program implementation [88,89]. Prior studies indicated that several measures contribute to the improvement in adherence, including telephone contact with a navigator, narrative invitation letters, and an approach in which the awareness of CRC and of purpose of CRC screening is strengthened by using an enhanced procedural informational brochure [90,91,92,93]. For screening initiation age, a prior study also revealed that the effectiveness of CRC screening was influenced [73]. American Cancer Society recommends that starting CRC screening at age 45 instead of 50 leads to more favorable cost-effectiveness [31,73]. Our finding also suggested that younger screening initiation ages are correlated with higher CRC-specific MR. Because we did not consider screening costs, we can, however, not conclude early initiation ages are dominant scenarios.

An explanation for the publication bias in model studies is that screening techniques are sensitive to early cancers and precursor lesions, which could be detected and treated at early stage [4,8]. The majority of results are positive due to early diagnosis, which does not introduce bias into our results.

### Strengths and Limitations

This study combined results reported in the latest English publication of CRC screening RCTs worldwide, which is the best representative of CRC screening effectiveness. This is also the first study that pooled the benefits of CRC screening in model studies and compared the outcomes with RCTs. Additionally, this study reviewed the effects of CRC screening on all-cause MR. There are some limitations in our study. First, populations in both RCT and model studies were only from Europe, the United States and Australia, so generalization of findings to other parts of the world should be carried out with caution. Second, this study did not include cost, detection rate, and false positive rates, which need to be considered when evaluating optimal screening scenarios and should therefore be added to future research. Third, the assumptions and parameters of the simulation models differed, which leads to a variability in results. There were relatively more publications on CRC-SPIN, SimCRC, and MISCAN-Colon, which are the three main models recommended by CISNET. Thus, the effects of these three models might have a greater impact on the pooled estimates compared to the other models. This resulted inevitably in a quasi-publication bias. Fourth, this study focused on commonly used scenarios. Further systematic reviews and meta-analyses focusing on scenarios with other ages, intervals, and novel screening techniques are necessary to expand the scope of screening effectiveness assessment. Fifth, only perfect adherence and widely used screening ages were considered in the model studies. Screening interval, age of screening initiation, and adherence, which might influence screening effectiveness, were included in scenario construction or sensitivity analyses in some model studies. However, there are no studies that explored MR as a function of different screening scenarios and adherence.

## 5. Conclusions

Our systematic review and meta-analysis provides a summary of the latest RCT and model studies of CRC screening on all-cause and CRC-specific MR. Commonly adopted global and Dutch screening scenarios could decrease CRC-specific mortality significantly, and 10-yearly colonoscopy screening is likely to be the most effective. Compliance-adjusted outcome with gFOBT in RCTs showed 1% of all-cause MR, and 0.62–1.1% reductions were shown in model studies with single FS screenings. Our findings suggest that adherence is an important factor in CRC-specific mortality and is an explanation for discrepancy in pooled estimates. Therefore, increased CRC screening adherence improves screening effectiveness. In model studies real-life adherence data should be used, and external validation should be performed for realistic outcomes. Lower screening initiation ages reduces CRC mortality.

## Figures and Tables

**Figure 1 cancers-15-01948-f001:**
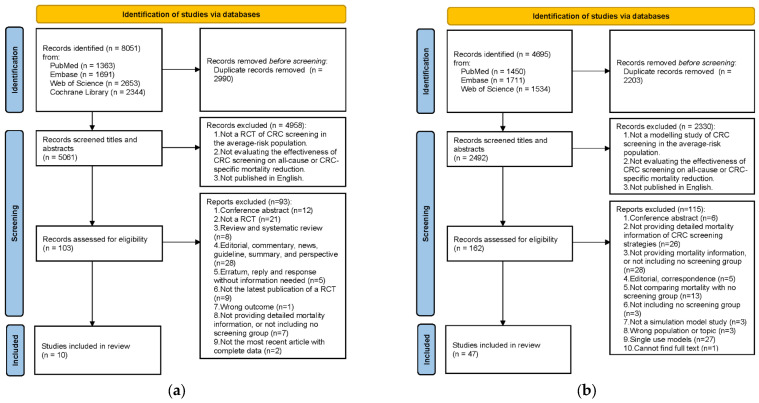
Flowchart of the selection process. (**a**) Studies on RCTs; (**b**) Studies on simulation model studies.

**Figure 2 cancers-15-01948-f002:**
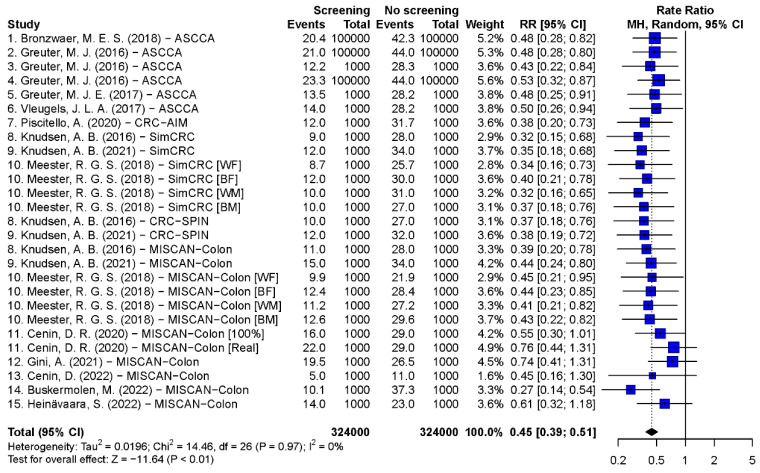
Forest plots of the CRC-specific mortality rate ratio on biennial FIT screening from the age of 55 to 75 [12,19,22,26,27,28,30,31,36,40,43,59,60,62,63].

**Figure 3 cancers-15-01948-f003:**
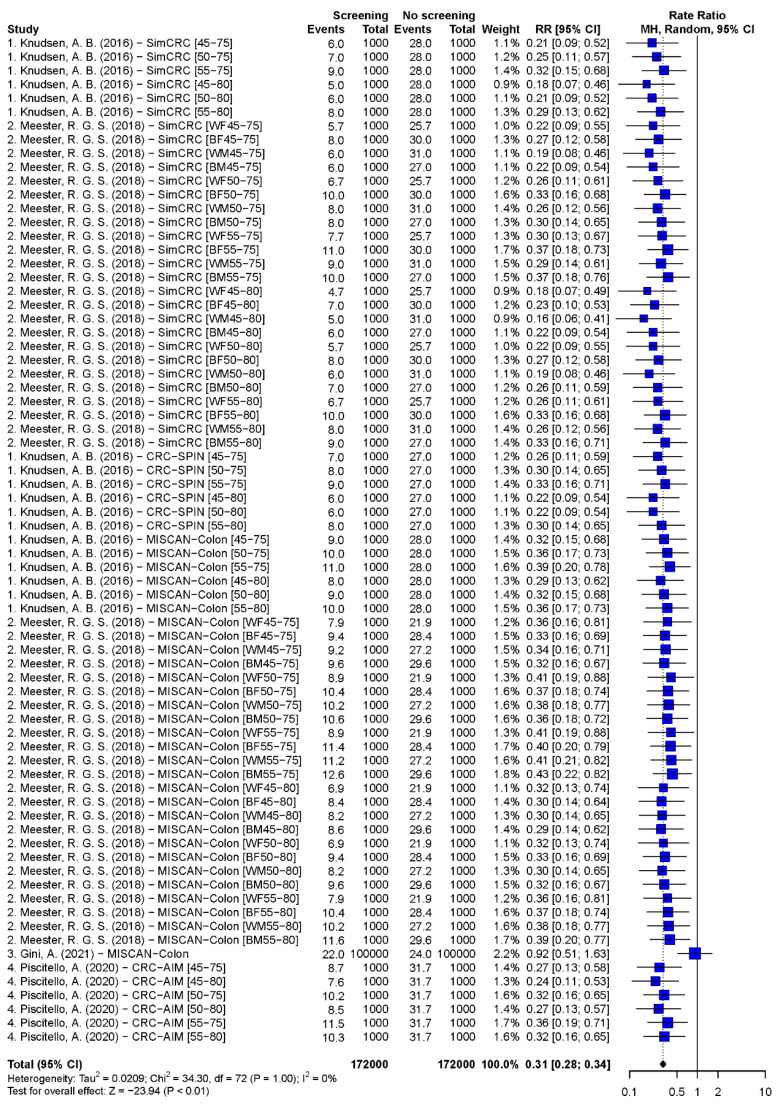
Forest plots of the CRC-specific mortality rate ratio on biennial gFOBT screening from the age of 45 to 80 [25,31,36,40].

**Figure 4 cancers-15-01948-f004:**
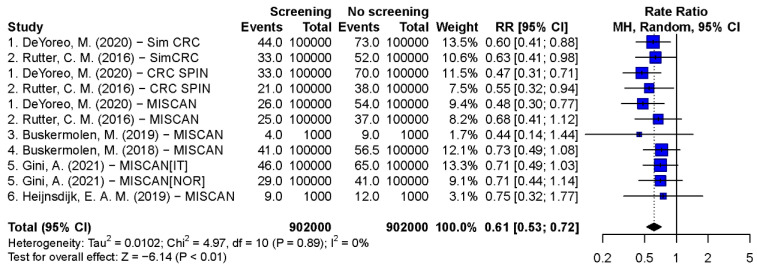
Forest plots of the CRC-specific mortality rate ratio on single FS screening from the age of 50 to 75 [20,21,24,25,29,42].

**Table 1 cancers-15-01948-t001:** Characteristics of randomized controlled trials.

Study	Program	Intervention	Age	Screening Years	Follow-Up Time (Years)	Total No. of Participants	Total No. of Screening Group	Adherence Rate in Screening Group (%)	Total No. of Control Group	Results of Compliance-Adjusted
A Shaukat US, 2013 [52]	Minnesota Colon Cancer Control Study	Annual or biennial gFOBT	50–80	1976–1992	30.0	46,551	31,157	90.0	15,394	All-cause and CRC-specific
A Shaukat Finland, 2021 [51]	Funen Fecal Occult Blood Trial	Biennial gFOBT	45–75	1985–2002	30.0	61,933	30,966	66.8	30,964	All-cause and CRC-specific
E Lindholm Sweden, 2008 [54]	Goteborg FOBT Trial	2–3 times gFOBT	60–64	1982–1995	Mean 15.5/8.7 ^a^	68,308	34,144	70.0	34,164	N/A
J H Scholefield UK, 2012 [49]	Nottingham Trial	Biennial gFOBT	45–74	1981–1991	Median 19.5	151,975	76,056	57.0	75,919	CRC-specific
J Pitkäniemi Finland, 2015 [56]	Finnish FOBT Screening Programme	Biennial gFOBT	60–69	2004–2012	Median 4.5	360,492	180,210	68.8	180,282	N/A
E Thiis-Evensen Norway, 2013 [57]	Telemark Polyp Study	Single FS	50–59	1983–1996	26.0	799	400	81.0	399	N/A
C Senore Italy, 2022 [50]	SCORE Trial	Single FS	55–64	1995–1999	Median 18.8	34,272	17,136	57.8	17,136	All-cause and CRC-specific
Ø Holme Norway, 2018 [53]	NORCCAP	Single FS with or without FOBT	50–64	1999–2001	Median 14.8	98,678	20,552	63.1	78,126	N/A
P F Pinsky US, 2019 [55]	PLCO Cancer Screening	FS at baseline and at year 3 or 5	55–74	1993–2001	Median 17.0	154,887	77,443	86.6	77,444	N/A
W Atkin UK, 2017 [48]	UKFSST	Single FS	55–64	1994–1999	Median 17.0	170,034	57,098	71.1	112,936	All-cause and CRC-specific

^a^ 15.5 years from the first invitation, 8.6 years from the last screening occasion. N/A: Not available. The study did not provide the result.

**Table 2 cancers-15-01948-t002:** Overview of randomized controlled trials on impact of CRC screening: results on all-cause and CRC-specific mortality (Intention-to-screen analyses ^a^).

Studies	Screening Group			Control Group			All-Cause Mortality Rate Ratio (95 CI%) ^b^	CRC-Specific Mortality Rate Ratio (95 CI%) ^b^
	All-Cause Deaths	CRC-Specific Deaths	Total Person-Years of Observation	All-Cause Deaths	CRC-Specific Deaths	Total Person-Years of Observation		
FOBT								
A Shaukat 2013 [52]	22,076	437	951,047	10,944	295	469,897	1.00 (0.97, 1.02)	0.73 (0.63, 0.85)
A Shaukat 2021 [51]	22,474	786	605,023	22,535	851	603,953	1.00 (0.98, 1.01)	0.92 (0.84, 1.02)
E Lindholm 2008 [54]	10,591	252	471,072	10,432	300	471,980	1.02 (0.99, 1.04)	0.84 (0.71, 1.00)
J H Scholefield 2012 [49]	40,681	1176	1,296,712	40,550	1300	1,296,614	1.00 (0.99, 1.02)	0.90 (0.84, 0.98)
J Pitkäniemi 2015 [56]	8000	170	805,480	7963	164	805,693	1.00 (0.97, 1.04)	1.04 (0.84, 1.28)
Meta-analysis:								
No. of studies							5	5
Pooled estimate							1.00 (0.99, 1.01)	0.88 (0.80, 0.96)
Test for overall effect: *P*							0.65	0.005
Heterogeneity: *I^2^* (%)*/Tau^2^/P*							0/0.00/0.74	59/0.01/0.04
FS								
E Thiis-Evensen 2013 [57]	188	1	8441	151	7	8997	1.33 (1.07, 1.64)	0.15 (0.02, 1.24)
C Senore 2022 [50]	3062	122	296,730	3155	157	295,013	0.96 (0.92, 1.01)	0.77 (0.61, 0.98)
P F Pinsky 2019 [55]	22,562	416	1,234,900	22,652	546	1,222,450	0.99 (0.97, 1.00)	0.75 (0.66, 0.86)
W Atkin 2017 [48]	13,279	353	902,198	26,409	996	1,780,738	0.99 (0.97, 1.01)	0.70 (0.62, 0.79)
Ø Holme 2018 [53]	3809	122	291,075	13,433	530	1,114,581	1.09 (1.05, 1.13)	0.88 (0.72, 1.07)
Meta-analysis:								
No. of studies							5	5
Pooled estimate							1.02 (0.97, 1.06)	0.76 (0.68, 0.84)
Test for overall effect: *P*							0.46	<0.001
Heterogeneity: *I^2^* (%)/*Tau^2^*/*P*							88/0.00/<0.001	35/0.00/0.19

^a^ Intention-to-screen analyses: including all individuals as randomized. Not be adjusted by compliance. ^b^ Control group was used as the reference.

**Table 3 cancers-15-01948-t003:** Characteristics of simulation models on biennial FIT screening from the age of 55 to 75.

Model	Study No.	Study	Simulate Population	Screening Age	Simulation Period/Follow-Up Time (Years)	Adherence Rate in Screening Group (%)	CRC Mortality of No Screening Group	CRC Mortality of Screening Group
ASCCA	1	Bronzwaer, M. E. S. (2018) [19]	Dutch	55–75	2014–2044	73% (FIT)/92% (FIT-positive CS)	42.3/100,000	20.4/100,000
	2	Greuter, M. J. (2016) [26]	Dutch	55–75	2014–2044 (30 years)	63% (FIT)/82% (FIT-positive CS)	44.0/100,000	21.0/100,000
	3	Greuter, M. J. (2016) [27]	Dutch	55–75	Lifetime (11 rounds)	63% (FIT)/96% (FIT-positive CS)	28.3/1000	12.2/1000
	4	Greuter, M. J. (2016) [28]	Dutch	55–75	2014–2044 (30 years)	63% (FIT)/96% (FIT-positive CS)	44.0/100,000	23.3/100,000
	5	Greuter, M. J. E. (2017) [12]	Dutch	55–75	Lifetime	72.6% (FIT)/92% (FIT-positive and Surveillance CS)	28.2/1000	13.5/1000
	6	Vleugels, J. L. A. (2017) [43]	Dutch	55–75	Lifetime	73% (FIT)/92% (FIT-positive CS)	28.2/1000	14.0/1000
CRC-AIM	7	Piscitello, A. (2020) [40]	US	55–75	Lifetime	100%	31.7/1000	12.0/1000
SimCRC	8	Knudsen, A. B. (2016) [31]	US	55–75	Lifetime	100%	28.0/1000	9.0/1000
	9	Knudsen, A. B. (2021) [30]	US	55–75	Lifetime	100%	34.0/1000	12.0/1000
	10	Meester, R. G. S. (2018) [36]	US White Female	55–75	Lifetime	100%	25.7/1000	8.7/1000
			US Black Female	55–75	Lifetime	100%	30.0/1000	12.0/1000
			US White Male	55–75	Lifetime	100%	31.0/1000	10.0/1000
			US Black Male	55–75	Lifetime	100%	27.0/1000	10.0/1000
CRC-SPIN	8	Knudsen, A. B. (2016) [31]	US	55–75	Lifetime	100%	27.0/1000	10.0/1000
	9	Knudsen, A. B. (2021) [30]	US	55–75	Lifetime	100%	32.0/1000	12.0/1000
MISCAN-Colon	8	Knudsen, A. B. (2016) [31]	US	55–75	Lifetime	100%	28.0/1000	11.0/1000
	9	Knudsen, A. B. (2021) [30]	US	55–75	Lifetime	100%	34.0/1000	15.0/1000
	10	Meester, R. G. S. (2018) [36]	US White Female	55–75	Lifetime	100%	21.9/1000	9.9/1000
			US Black Female	55–75	Lifetime	100%	28.4/1000	12.4/1000
			US White Male	55–75	Lifetime	100%	27.2/1000	11.2/1000
			US Black Male	55–75	Lifetime	100%	29.6/1000	12.6/1000
	11	Cenin, D. R. (2020) [22]	Australian	54–74	Lifetime 40–100	100%	29.0/1000	16.0/1000
			Australian	54–74	Lifetime 40–100	Realistic adherence	29.0/1000	22.0/1000
	12	Gini, A. (2021) [62]	Dutch	55–75	2018–2050	71.3%	26.5/1000	19.5/1000
	13	Cenin, D. (2022) [60]	Chinese	55–75	Lifetime	100%	11.0/1000	5.0/1000
	14	Buskermolen, M. (2022) [59]	Dutch	55–75	Lifetime	72.4%	37.3/1000	10.1/1000
	15	Heinävaara, S. (2022) [63]	Finnish	55–74	Lifetime	100%	23.0/1000	14.0/1000

## Data Availability

Data are contained within the article or Appendix A.

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
