# Peer review of "Effectiveness of Colorectal Cancer (CRC) Screening on All-Cause and CRC-Specific Mortality Reduction: A Systematic Review and Meta-Analysis"

_cancers, 2023, doi:10.3390/cancers15071948_

Round 1

Reviewer 1 Report

Zheng el al. wrote a systematic review on “Effectiveness of colorectal cancer (CRC) screening on all-cause and CRC-specific mortality reduction: A systematic review and meta-analysis” that provides a comprehensive review of randomized control trials (RCTs) and simulation models that estimate the effectiveness of different screening methods for colorectal cancer (CRC) mortality reduction. The authors reported that biennial FIT, gFOBT, single/5 yearly FS, and 10-yearly colonoscopy screenings significantly reduced CRC-specific mortality, with 10-yearly colonoscopy being the most effective with a mortality reduction of 73%. The study also highlights the importance of adherence as a factor in CRC-specific mortality reduction, which may explain the discrepancy in pooled estimates reported in the literature thus far. The authors provided evidence that adherence rates affect the effectiveness of gFOBT screening, with higher adherence rates leading to higher mortality reduction.

Overall, the manuscript is a valuable contribution to the literature on CRC screening effectiveness and mortality reduction. However, the manuscript could be improved by addressing the following points:

1.     The introduction lacks a clear research question or objective and could provide more context on the burden of CRC and the current guidelines for screening.

2.     The description of the search strategy is not clear. The authors should provide more details about the search terms and search strings used in each database, not only PubMed. The methods section could be more detailed in describing the search strategy. This will help readers evaluate the validity and reliability of the study.

3.     The authors should mention the inclusion/exclusion criteria in the main text instead of referring to Supplementary materials, Table S1.

4.     The authors should provide more information on the adherence to screening, as this is an important factor in the effectiveness of CRC screening programs.

5.     Clarification is needed on whether the compliance-adjusted analysis was performed for both RCTs and model studies or only for RCTs.

6.     More details on the statistical methods used for the meta-analysis should be provided. Discussing why a particular analysis tool was chosen is very important to validate the analysis.

7.     Overall results presentation is fine and sufficiently describes. However, Figure 1 does not follow the PRISMA prescribed template. Authors need to present the information correctly.

8.     The discussion highlights the importance of adherence in the effectiveness of CRC screening, but it could be expanded to discuss strategies to improve adherence.

9.     The discussion section could provide a more in-depth analysis of the limitations of the study and suggestions for future research.

10.  The manuscript contains some grammatical errors and should be proofread and edited for concise and specific language.

Reviewer 2 Report

thank you for the opportunity to review this literature review. this review highlights an already known notion that screening can only be effective if participation is comprehensive. among the included studies using statistical models, are there any data that have modelled the reduction in cancer-specific mortality as a function of variability in screening? 

the authors suggest that early screening would further reduce colorectal cancer-specific mortality. among the included studies using statistical models, are there data that model the reduction in cancer-specific mortality as a function of age of screening initiation? 

Round 2

Reviewer 1 Report

Authors addressed all the review comments.